# [Re] AdaBelief Optimizer: Adapting Stepsizes by the Belief in Observed Gradients

## Reproducibility Summary

**Scope of Reproducibility**

The proposed optimizer: AdaBelief, claims to achieve three goals: fast convergence as in adaptive methods, good generalization as in SGD, and training stability. We perform experiments to validate the claims of the paper [28].

**Methodology**

To validate these claims, we reproduce experiments on **Image Classification** with CIFAR-10, CIFAR-100 and ImageNet datasets, **Language Modeling** with Penn Treebank, **Generative Modeling** with WGAN, WGAN-GP and SN-GAN architectures. We use the code provided by the author[1]. All experiments were performed on 8 NVIDIA V100 GPUs and took about 1096 GPU hours in total. Our entire code is provided in the supplementary material.

**Results**

The image classification experiments on CIFAR-10, CIFAR-100 and ImageNet are reproduced to within 0.29%, 0.18% and 0.25% of reported values respectively. The language modeling experiments produce an average deviation of 0.22%, while the generative modeling experiments on WGAN, WGAN-GP and SN-GAN are replicated to within 2.2%, 1.8% and 0.33% of reported value.

We perform ablation studies for change of dataset in language modeling and for effect of weight decay on ImageNet. We also perform analysis of generalization ability of optimizers and of training stability of GANs. All of the results largely support the claims made in the paper [28].

**What was easy**

The authors provide implementation for most of the experiments presented in the paper. Well documented code and lucid paper helped understand the experiments clearly.

**What was difficult**

The challenging aspects in our study were: (1) Grid search for optimal hyperparameters (HP) in cases where HP were not provided or results did not match, (2) time and resource intensive experiments like ImageNet ($\sim$ 22 hrs.) and SN-GAN ($\sim$ 15 hrs.), (3) writing code to evaluate claims of the AdaBelief paper.

**Communication with original authors**

We communicated the original author Juntang Zhuang on multiple occasions for doubts related to hyperparameters and code, to which he promptly replied and helped us.

---

[1]https://github.com/juntang-zhuang/Adabelief-Optimizer

# 1  Introduction

Optimization is at the heart of machine learning. Training of neural networks aims to find the optimal solution (deepest valley on the loss surface) using gradient descent. The variation in method to traverse the loss landscape gives rise to different optimizers. Discovering different optimizers is an active area of research in machine learning. In this report, we reproduce and add on to the experimental analysis of a recent optimizer, AdaBelief [28], introduced in 2020 at NeurIPS conference.

The proposed optimizer, AdaBelief, claims to outperform its counterparts on various real world deep learning tasks. As a part of the ML Reproducibility Challenge, we replicate all the experiments mentioned in the AdaBelief paper [28], comparing it with other optimizers, and also perform additional experiments to investigate the efficacy of AdaBelief.

# 2  Details of Optimizers

Optimizers are of two types: **(1) accelerated Stochastic Gradient Descent (SGD) family** [24] that includes SGD with momentum [26] & Nestrov Accelerated Gradient (NAG) [22], and **(2) adaptive methods** like Adam [12], RAdam [13], AdamW [14], RMSProp [7], Yogi [27], AdaBound [15], AdaBelief [28], MSVAG [2], Fromage [3], Apollo [16].

SGD [24] family uses the same learning rate for all parameters, whereas, adaptive methods update their parameters as a function of gradients. While this has shown success in faster convergence due to a more streamlined trajectory, it has raised questions regarding the generalization ability of adaptive methods. RMSProp [7] builds over SGD by penalizing updates in directions that have high gradients. The intuition behind this is to prevent drastic updates in particular directions. It does so by damping magnitude of update by factor of exponential moving average (EMA) computed for squares of gradients. Adam [12] improves over RMSProp by introducing a momentum term that helps prevent over-damping of step size in case of RMSProp. RAdam [13] seeks to tackle the convergence problem of Adam by proposing to use a small learning rate during initial stages of training when variance is high, while AdamW [14] and MSVAG [2] address the generalization problem in Adam. AdamW does this by introducing a weight decay regularization term and MSVAG decomposes Adam as a sign update and magnitude scaling. Yogi [27] considers the effect of mini-batch size and proposes an update equation that has shown to outperform Adam with very little hyperparameter tuning. AdaBelief [28] amplifies (or dampens) its updates by a factor proportional to the 'belief' in observed gradient i.e. square of difference between the observed gradient and EMA of the gradient. AdaBound [15] bridges the gap between SGD family and Adaptive methods by making use of an update that smoothly transitions from Adam to SGD. Fromage [3] takes a different path to optimization - it accounts for the network structure by looping in weight matrices into the update equation. Apollo [16] takes a step forward from the aforementioned first order optimizers by approximating the Hessian via a diagonal matrix, keeping computations in-line with first-order schemes.

# 3  Scope of reproducibility

AdaBelief [28] claims to performs better than existing optimizers. To evaluate the validity of its claims, we investigate the following target questions:

- Does AdaBelief produce better scores in comparison to other optimizers on real world tasks of image classification, language modeling, generative modeling and reinforcement learning?
- Does AdaBelief convergence fast like adaptive methods, e.g. Adam?
- Does AdaBelief generalize well like the accelerated gradient methods, e.g. SGD?
- Adaptive methods like Adam are stable in complex settings like training of Generative Adversarial Networks (GANs) [6]. How does AdaBelief compare to them?

# 4  Methodology

## 4.1  Experimental setup and model description

We perform experiments on many real world tasks: **(a) Image Classification**: CIFAR-10, CIFAR-100 & ImageNet datasets are used. On CIFAR-10 & CIFAR-100, we train using VGG11 [25], ResNet34 [9] and DenseNet121 [11] . In the case of ImageNet we use a ResNet18 architecture. **(b) Language Modeling**: Penn Treebank [17] and WikiText-2 [18] datasets are used. Both are used to train 1, 2, 3-layer LSTM [10]. The HP of the LSTM model were taken

from here[2]. **(c) Generative Modeling**: CIFAR-10 dataset is used with Wasserstein-GAN (WGAN) [1], with the improved gradient penalty version WGAN-GP) [8] & with spectral normalization GAN (SN-GAN) [20] architectures, where generator and discriminator use same HP. WGAN is a smaller model with a vanilla CNN generator, whereas the SN-GAN is a bigger model with spectral normalization in the discriminator. For SN-GAN we make use of this repository[3] **(d) Reinforcement Learning**: An agent is trained by Adam and AdaBelief optimizers to play Space Invaders (Atari Game) using Deep Q-Network (DQN) [21] architecture. Implementation was taken from here[4]. The code for experiments on image classification, language modeling, WGAN, WGAN-GP was taken from here[5].

| S. No. | Task | Dataset | Setup | Rep. Status | Our Contribution | No. of Exp. | GPU HPR | Total GPU hours |
|---|---|---|---|---|---|---|---|---|
| 1. | Image Classification | CIFAR-10 | VGG, RN, DN | ✓ | Exp. on Apollo; bias-variance anal. | 30 | 2.5 | 75 |
| 2. | | CIFAR-100 | VGG, RN, DN | ✓ | Exp. on Apollo; bias-variance anal. | 30 | 2.5 | 75 |
| 3. | | ImageNet | ResNet18 | ✓ | Analysis of weight decay | 3 | 22 | 66 |
| 4. | Language Modeling | PTB, WT2 | LSTM (1 layer) | ✓ | Fromage LRS; WT2 | 11 | 1.33 | 14.63 |
| 5. | | | LSTM (2 layer) | ✓ | AdamW & RAdam LRS; WT2 | 11 | 2.5 | 27.5 |
| 6. | | | LSTM (3 layer) | ✓ | AdamW & RAdam LRS; WT2 | 11 | 3.75 | 41.25 |
| 7. | Generative Modeling | CIFAR-10 | WGAN [1] | ✓ | N/A (only reproduced paper's [28] exp.) | 70 | 0.89 | 53.55 |
| 8. | | | WGAN-GP [8] | ✓ | N/A (only reproduced paper's [28] exp.) | 70 | 1 | 66.5 |
| 9. | | | SN-GAN [20] | ✓ | HP search; training stablity anal. | 45 | 15 | 675 |
| 10. | Reinforcement Learning | N/A | Space Invaders (Atari) | ✓ | Beyond AdaBelief paper [28] | 2 | 1 | 2 |

Table 1: Summary of our contributions and reproducibilty details of performed experiments. Exp. 1 to 9 are mentioned in the AdaBelief paper [28] and have been reproduced successfully along with some additional contribution to each experiment. We also perform exp. 10 which is not a part of AdaBelief paper. [**Legend** - **Rep.**: Reproducibility, **Exp.**: "Experiment(s)", **HPR**: "hours per run", **RPO**: "runs per optimizer", **anal.**: "analysis", **HP**: "hyperparameter", **LRS**: "Learning Rate Search", **WT2**: "WikiText-2", **DN**: "DenseNet121", **RN**: "ResNet34", **VGG**: "VGG11", **PTB**: "Penn Treebank"]

## 4.2 Datasets

The following datasets were used in the experiments - **(a) CIFAR-10**: It consists of $60,000$ images of size $32 \times 32$, grouped into 10 classes (6000 images per class). We use the default train-test split of $50,000 : 10,000$. **(b) CIFAR-100**: It is same as CIFAR-10 but the images are grouped into 100 classes (600 images per class). **(c) ImageNet** [5]: We use ILSVRC 2012 dataset[6] which consists of $\sim 1.35M$ images of size $256 \times 256$ split into 1000 classes. Train-val-test split is $1,281,167 : 50,000 : 100,000$. As part of pre-processing we remove mis-labelled data[7] **(d) Penn Treebank**[8] (PTB) [17]: The train-val-test split of tokens is $887,521 : 70,390 : 78,669$. **(e) WikiText-2** (WT2) [18]: It is a subset of WikiText-103, features a larger vocabulary and retains the punctuation, original case and numbers which are omitted in PTB dataset. We ran experiments on WT2[9] using the train-val-test token split of $2,045,059 : 213,119 : 240,498$.

## 4.3 Hyperparameters

In this section we mention the HP used by optimizers in our experiments. Optimal values of commonly used HP are listed in Table 2. Below we mention the source of these values and details of HP search.

For most experiments, we use the optimizer-specific HP as mentioned in the original repository[5] since searching the HP for all experiments is computationally infeasible. However, the repository does not mention the HP for SN-GAN & Fromage, and the mentioned HP for 2- and 3-layer AdamW & RAdam resulted in large deviation. So, we perform learning rate (LR) search for **Fromage** and 2- & 3-layer **AdamW** and **RAdam** over the interval $[10^{-3}, 10^{-2}]$ (5 values). For **SN-GAN**, we search $\beta_1$ (3 values in $[0.4, 0.9]$) and $\epsilon$ (3 values in $[10^{-12}, 10^{-6}]$). For **Reinforcement Learning**, we use LR of $10^{-4}$ and $\epsilon = 10^{-10}$ for AdaBelief and Adam, as mentioned on the RL repository[4].

---

[2]https://github.com/salesforce/awd-lstm-lm

[3]https://github.com/juntang-zhuang/SNGAN-AdaBelief

[4]https://github.com/juntang-zhuang/rainbow-adabelief

[5]https://github.com/juntang-zhuang/Adabelief-Optimizer

[6]ImageNet dataset (Kaggle)

[7]Blacklisted images (GitHub)

[8]Penn Treebank Dataset

[9]WikiText-2 dataset

Now we list the HP which are specific to each optimizer. The LR decays to $1/10^{th}$ of its value at $150^{th}$ epoch for image classification on CIFAR-10 and CIFAR-100, and at epoch 70 & 80 on ImageNet. **AdaBelief** uses `weight_decouple=False, fixed_decay=False, rectify=False` for all the experiments and `weight_decouple=True` on ImageNet. **SGD** uses `momentum=0.9`, and **Apollo** uses `warmup=200`, `weight_decay_type='L2'` for image classification on CIFAR-10 and CIFAR-100. **AdaBound** uses `final_lr=30` on PTB and `final_lr=0.01` with GAN experiments.

| Task | Setup | Learning Rate | $\beta_1$ | $\beta_2$ | $\epsilon$ | Weight Decay | Epochs |
|---|---|---|---|---|---|---|---|
| Image Classification | CIFAR | $10^{-3}$ $(10^{-1}_{S,M}, 1_L)$ | 0.9 | 0.999 | $10^{-8}$ $(10^{-3}_Y, 10^{-4}_L)$ | $5 \times 10^{-4}$ $(10^{-2}_W, 2.5 \times 10^{-4}_L)$ | 200 |
| | ImageNet | $10^{-3}$ | 0.9 | 0.999 | $10^{-8}$ | $10^{-2}$ | 90 |
| Language Modeling | 1 layer | $10^{-3}$ $(30_{S,M}, 10^{-2}_{Y,D,F})$ | 0.9 | 0.999 | $10^{-8}$ $(10^{-16}_B, 10^{-3}_Y)$ | $1.2 \times 10^{-6}$ | 200 |
| | 2 layer | $10^{-2}$ $(30_{S,M}, 10^{-3}_{W,R})$ | 0.9 | 0.999 | $10^{-8}$ $(10^{-12}_B, 10^{-3}_Y)$ | $1.2 \times 10^{-6}$ | 200 |
| | 3 layer | $10^{-2}$ $(30_{S,M}, 10^{-3}_{W,R})$ | 0.9 | 0.999 | $10^{-8}$ $(10^{-12}_B, 10^{-3}_Y)$ | $1.2 \times 10^{-6}$ | 200 |
| Generative Modeling | WGAN | $2 \times 10^{-4}$ | 0.5 | 0.999 | $10^{-8}$ $(10^{-12}_B)$ | $0$ $(5 \times 10^{-4}_P)$ | 100 |
| | WGAN-GP | $2 \times 10^{-4}$ | 0.5 | 0.999 | $10^{-8}$ $(10^{-12}_B)$ | $0$ $(5 \times 10^{-4}_P)$ | 100 |
| | SN-GAN | $2 \times 10^{-4}$ | 0.5 | 0.999 | $10^{-8}$ $(10^{-6}_A, 10^{-12}_B)$ | $0$ | 100000 |

Table 2: Optimizer specific hyperparameter (HP) values and epochs for experiments performed. Each cell follows a format $X(Y)$ where $X$ is the optimal value of the HP unless stated otherwise and $Y$ contains elements of the form $v_o$ where $v$ is the value of HP for optimizer $o$. The abbreviations used for optimizers are (**S**)GD, (**A**)dam, Adam(**W**), Ada(**B**)elief, (**Y**)ogi, (**M**)SVAG, (**R**)Adam, (**F**)romage, AdaBoun(**D**), Apo(**L**)lo, (**P**)adam

## 4.4 Computational requirements

We run experiments on a Portable Batch System (PBS) managed cluster. We used 8 NVIDIA V100 GPUs and 384 GB RAM. All experiments except ImageNet use a single GPU. GPU runtime of all experiments are listed in table 1.

# 5 Experiments and Results

## 5.1 Experiments reproducing original paper

To evaluate the performance of AdaBelief and to validate the aforesaid claims, we perform experiments on various tasks like Image Classification, Language Modeling, Generative Modeling, Reinforcement Learning and compare our results with those stated in the paper [28]. HP details can be found in Table 2

### 5.1.1 Image classification

We run experiments on CIFAR-10 and CIFAR-100 using VGG11 [25], Resnet34 [9] and DenseNet121 [11] architectures, performimg 3 independent runs on 9 optimizers[10]. Additionally, we perform experiments using Apollo optimizer [16], that has claimed to outperform AdaBelief on CIFAR datasets with ResNet110 architecture. Fig. 1 plots test accuracy results. Plots for train accuracies are in supplementary (Fig. 5). All the obtained results agree with those reported in the AdaBelief paper [28].

To assess the performance on large scale datasets, we ran experiments on ImageNet [5]. We follow a similar setting as the author and run experiments on AdaBelief [28] and MSVAG [2] and report results for remaining optimizers from literature (Table 3). The top-1 accuracy lags by 0.32% and 0.18% respectively in case of AdaBelief and MSVAG. Other optimizers from literature use weight decay of $10^{-4}$ while the author performs experiments on AdaBelief using a value of $10^{-2}$. We analyse the effect of weight decay in section 6.2.

| Adabelief | SGD | Adabound | Yogi | Adam | MSVAG | RAdam | AdamW |
|---|---|---|---|---|---|---|---|
| 69.76 | **70.23**[†] | 68.13[†] | 68.23[†] | 63.79[†] (66.54[‡]) | 65.81 | 67.62[‡] | 67.93[†] |

Table 3: Top-1 accuracy of ResNet18 on ImageNet. † is reported in [4], and ‡ is reported in [13]

---

[10]SGD, Adam, AdamW, AdaBelief, Yogi, MSVAG, RAdam, Fromage, AdaBound

 **5.1.2   Language Modeling**

We ran experiments on Penn Treebank (PTB) dataset [17] using 1,2,3-layer LSTM models. We report test perplexities (ppl) (Fig. 2) for 3 independent runs on 9 optimizers[10]. Plots for train ppl are in supplementary (Fig. 1). For Fromage, the author does not provide HP, hence we grid search and find the optimal $LR = 10^{-2}$. In case of 2 layer LSTM using AdamW & RAdam, we find that an $LR = 10^{-3}$ gives a ppl of 73.78 & 74.05, while $LR = 10^{-2}$ gives a ppl of 93.61 & 90.49 respectively. The author reports a ppl $\sim 73$, $\sim 73.5$ at $LR = 10^{-2}$. Similarly, in 3-layer LSTM, $LR = 10^{-3}$ for AdamW and RAdam works better than $LR = 10^{-2}$. PTB is a small dataset, so, we additionally experiment on WikiText-2 (section 6.1) for Adam and AdaBelief (top performers in case of PTB) on the setting reported here[11].

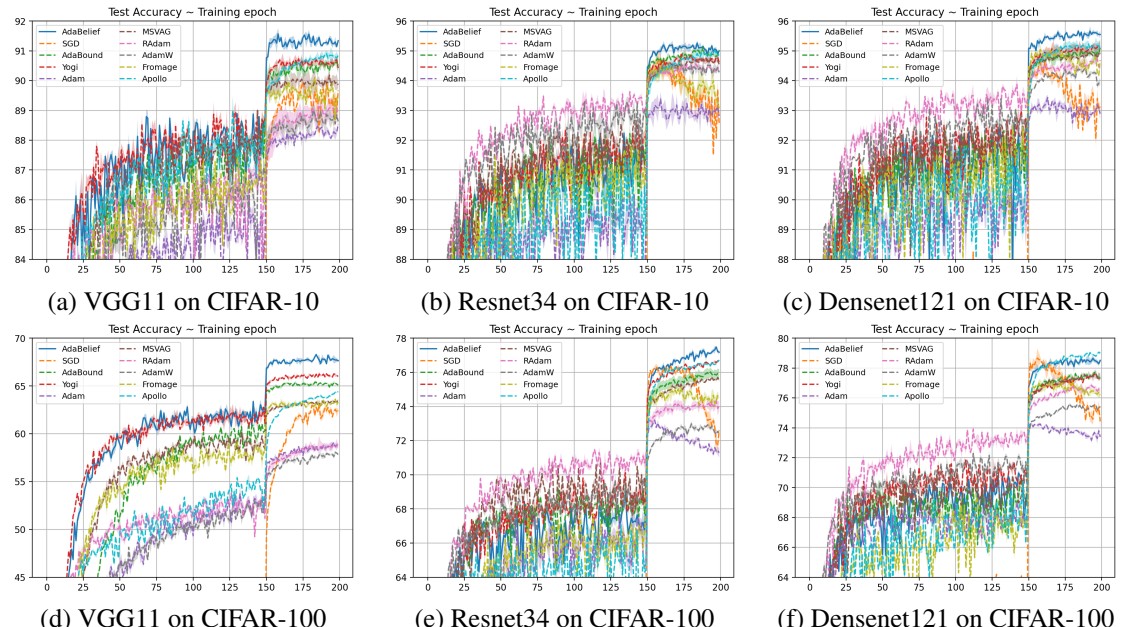

(a) VGG11 on CIFAR-10          (b) Resnet34 on CIFAR-10          (c) Densenet121 on CIFAR-10

(d) VGG11 on CIFAR-100          (e) Resnet34 on CIFAR-100          (f) Densenet121 on CIFAR-100

Figure 1: Test accuracy ($[\mu \pm \sigma]$) on CIFAR-10 and CIFAR-100

| Adabelief | RAdam | RMSProp | Adam | Fromage | Yogi | SGD | MSVAG | AdaBound |
|---|---|---|---|---|---|---|---|---|
| $12.98 \pm 0.22$ | $13.10 \pm 0.20$ | $\mathbf{12.86 \pm 0.08}$ | $13.01 \pm 0.15$ | $46.31 \pm 0.86$ | $14.16 \pm 0.05$ | $48.94 \pm 2.88$ | $56.89 \pm 2.61$ | $16.84 \pm 0.10$ |

Table 4: FID ($[\mu \pm \sigma]$) of a SN-GAN with ResNet generator on CIFAR-10.

**5.1.3   Generative Modeling**

We run experiments on WGAN [1], WGAN-GP [8] & SN-GAN [20]. SN-GAN makes use of a ResNet generator with spectral normalization in the discriminator and is trained for 100,000 steps. Five independent runs on 9 optimizers[12] are performed. We also perform these experiments using the Padam [19] optimizer on WGAN and WGAN-GP. FID values for SN-GAN and Padam (Table 4, 5). Fig. 4 shows the variation in FID during training, giving an idea of stability and convergence of different optimizers. Boxplots of FID values corresponding to multiple runs on WGAN and WGAN-GP are shown in Fig. 3. Collages of generated images for all optimizers can be found in supplementary (Fig. 7, 8, 9).

**(a) SN-GAN**: In case of Fromage [3] and MSVAG [2], we obtain $\sim 4$ and $\sim 8$ worse FID than what is reported, while for AdaBound [15] we obtain a $\sim 40$ better FID. We suspect the reason for this large deviation to be a difference in HP value being used. Since we performed a HP search for SN-GAN, our HP (Table 2) are optimal. The results of remaining optimizers were comparable to what was reported in the paper. **(b) WGAN:** We observe that AdaBelief outperforms other optimizers with a median FID of $\sim 80$ which agrees with reported value. We observe a significantly worse FID with Fromage. **(c) WGAN-GP:** AdaBelief and AdaBound achieve comparable results $\sim 67$ FID which are

---

[11]https://github.com/salesforce/awd-lstm-lm
[12]SGD, Adam, RMSProp, AdaBelief, Yogi, MSVAG, RAdam, Fromage, AdaBound

better than the other optimizers. Fromage shows similar deviation like in WGAN. With Padam, we find that for both WGAN and WGAN-GP, increasing the partial ($p$) i.e. moving from SGD towards Adam, decreases the FID. The FIDs obtained are found to agree with or are marginally better than what was stated in the paper.

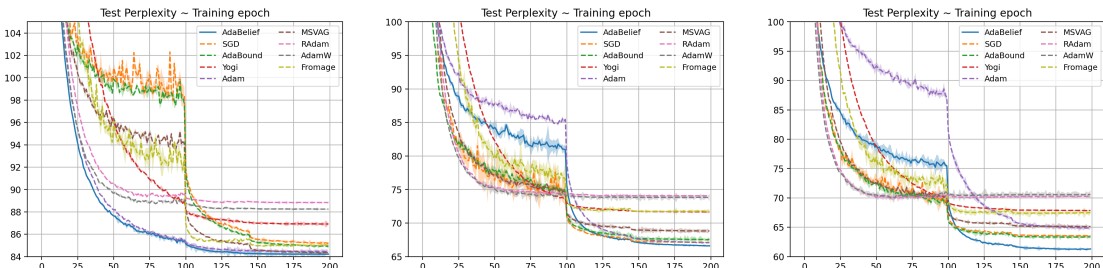

Figure 2: Left to right: Test perplexity ($[\mu \pm \sigma]$) on Penn Treebank for 1,2,3-layer LSTM

| | AdaBelief | Padam | | | | | | |
|---|---|---|---|---|---|---|---|---|
| | | p=1/2 (Adam) | p=2/5 | p=1/4 | p=1/5 | p=1/8 | p=1/16 | p=0 (SGD) |
| FID (WGAN) | $\mathbf{82.85 \pm 2.21}$ | $106.38 \pm 9.76$ | $95.66 \pm 3.76$ | $422.62 \pm 35.68$ | $396.69 \pm 24.91$ | $330.44 \pm 26.62$ | $357.26 \pm 32.39$ | $459.01 \pm 14.62$ |
| FID (WGAN-GP) | $75.37 \pm 7.37$ | $\mathbf{71.87 \pm 0.83}$ | $85.42 \pm 5.15$ | $152.34 \pm 17.49$ | $170.80 \pm 20.43$ | $205.57 \pm 13.79$ | $228.40 \pm 18.24$ | $236.99 \pm 7.26$ |

Table 5: FID values ($[\mu \pm \sigma]$) using AdaBelief and Padam on WGAN and WGAN-GP, Lower FID is better.

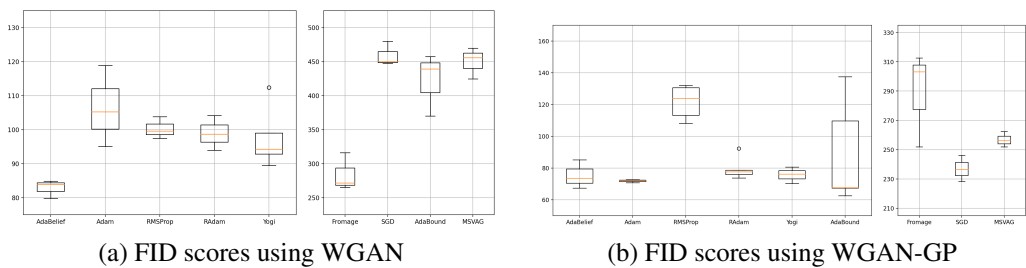

(a) FID scores using WGAN                (b) FID scores using WGAN-GP

Figure 3: FID score of WGAN and WGAN-GP using a vanilla CNN generator on CIFAR-10. Lower is better. For each model, successful and failed optimizers (i.e. ones with higher FID values) are shown in the left and right respectively, with different y-axis ranges.

## 5.2 Experiments beyond original paper

### 5.2.1 RL toy

To investigate the efficacy of AdaBelief in use cases beyond text and images we train an agent to play Space Invaders (Atari Game). We report $Q$ value and reward function for Adam and AdaBelief in supplementary (Fig. 10, 11). We compare our results with author's results from here[13] and find that both results agree.

### 5.2.2 Image Classification on CIFAR-10 and CIFAR-100 using Apollo

Apollo [16] is another optimizer that claims to achieve better convergence speed and generalization than SGD and variants of Adam. To investigate this, we experiment with Apollo on CIFAR-10 and CIFAR-100. Fig. 5, 6 (in supplementary) show the train, test accuracies on VGG11, ResNet34 and DenseNet121 for the 3 independent runs. AdaBelief outperforms Apollo in all settings except DenseNet121 on CIFAR-100 . It can also be seen that as we move from a simpler (VGG11) to a complex architecture (DenseNet121) the gap between Apollo and AdaBelief closes out. We made use of official implementation of Apollo in our experiments[14].

---

[13]https://github.com/juntang-zhuang/rainbow-adabelief

[14]https://github.com/XuezheMax/apollo

### 5.2.3 Evaluating GAN training stability

To assess stability of AdaBelief while training GANs, we look into difference between SN-GAN's generator and discriminator training losses on CIFAR-10. We do this for AdaBelief, Adam and RMSProp (since they have top-2 FID scores on SN-GAN) in the adaptive family, and with SGD for a comparison. Fig. 12 in supplementary plots the generator and discriminator training losses. We observe that the adaptive methods are more stable than SGD and within the adaptive family the order of stability from most stable to least stable varies as RMSProp, AdaBelief, Adam.

### 5.2.4 Evaluating generalization ability

To evaluate AdaBelief's ability to generalize, we analyze the bias and variance of image classification models trained using SGD, Adam, AdaBelief and Apollo optimizers on CIFAR-10 and CIFAR-100. We use the method outlined here [23] for bias-variance analysis. For each optimizer, we note its train and test accuracy (Fig. 1) corresponding to the epoch with best test accuracy (acc), and compute their difference. This data is stated as 3-tuples in Table 6. Lower training acc denotes high bias and vice-versa. The difference between the train and test acc is a measure of variance. Based on Table 6, we observe that AdaBelief models have the least bias on all configurations, while they have $2^{nd}$, $3^{rd}$ or $4^{th}$ lowest variance. SGD has the least variance on most configurations (highlighted in red), but their bias is high (mostly ranked $3^{rd}$ or $4^{th}$ in low bias).

| Optimizer | CIFAR-10 | | | CIFAR-100 | | |
|---|---|---|---|---|---|---|
| | VGG11 | ResNet34 | DenseNet121 | VGG11 | ResNet34 | DenseNet121 |
| SGD | 95.88, 89.95, 5.93 | 98.77, 94.72, 4.05 | 98.72, 94.61, 4.11 | 78.87, 63.09, 15.78 | 98.94, 76.35, 22.59 | 94.67, 78.67, 16.00 |
| Adam | 94.68, 88.54, 6.14 | 98.36, 93.38, 4.98 | 99.23, 93.43, 5.80 | 67.63, 59.08, 8.55 | 92.73, 73.20, 19.53 | 96.69, 74.28, 22.41 |
| AdaBelief | 99.36, 91.57, 7.79 | 99.96, 95.26, 4.70 | 99.97, 95.67, 4.30 | 98.84, 68.29, 30.55 | 99.97, 77.48, 22.49 | 99.96, 78.66, 21.30 |
| Apollo | 98.79, 90.91, 7.88 | 99.74, 95.01, 4.73 | 99.82, 95.23, 4.59 | 74.80, 64.42, 10.38 | 99.54, 76.72, 22.82 | 99.68, 79.06, 20.62 |

Table 6: Analysis of generalization capability of AdaBelief on CIFAR-10 and CIFAR-100 for VGG11, ResNet34 and DenseNet121 architectures using **bias** and **variance**. Each cell denotes a 3-tuple of the form (train acc, test acc, difference b/w train and test acc) corresponding to the model which achieves best test acc (out of 3 runs) for each configuration. For each column, the value in red denotes the optimizer with least **variance** (i.e. the least train-test acc difference) and the value in blue denotes the optimizer with least **bias** (i.e. with most training acc). AdaBelief models achieve the least bias on all configurations, while they lag behind in terms of variance.

### 5.2.5 Evaluating convergence speed

**Definition 5.1** (Epoch of Convergence (EC)). Let $m_k$ denote the metric (acc or ppl) at $k^{th}$ epoch. *EC* is then defined as the smallest epoch $x$ such that $|m_y - m_x| < \delta \ \forall \ y \in [x, \ x + w]$, where $w$ and $\delta$ are chosen as 15 and 0.05 respectively. In other words, *EC* is the smallest epoch for which there exists at least $w(= 15)$ epochs to its right with accuracies (or perplexities) within a fixed tolerance $\delta(= 0.05)$. If such $x$ cannot be found, the said optimizer is said to have *failed to converge (FTC)*.

To address the claim on convergence ability different optimizers (section 3) we make use of Def. 5.1. We perform the analysis for Image Classification and Language Modeling (section 5.1) experiments. We smoothen the accuracy (or perplexity) curves for all optimizers by finding the exponential moving average (EMA) with a smoothing factor $\beta = 0.7$. Analyzing the computed *ECs* yield that the convergence speed of AdaBelief is comparable to other members of Adaptive family for experiments performed on CIFAR datasets (Fig. 1). For Language Modeling experiments, we find that Adam and AdaBelief show similar convergence trends but considerably lag behind in comparison to RAdam, AdamW and Fromage (Fig. 2) that are unaffected by learning rate decay which takes place at $100^{th}$ epoch. For exact *EC* values refer Table 1 in supplementary.

## 6 Ablation studies

### 6.1 WikiText-2 on LSTM

To study the performance change due to a larger dataset, we ran Language Modeling experiments on WikiText-2 [18] using AdaBelief and Adam optimizers with 1, 2, 3 layer LSTM models. Fig. 3, 4 (in supplementary) show train and test perplexity for 3 independent runs. It can be seen that the performance of Adam and AdaBelief is comparable on 1 and 2 layer LSTM models, while in the 3 layer case AdaBelief outperforms Adam by $\sim 5$ ppl.

## 6.2 Effect of weight decay on ImageNet

The paper [28] uses a weight decay of $10^{-2}$ while experimenting with AdaBelief on ImageNet. However, the results for other optimizers are from the literature that typically use a (smaller) weight decay of $10^{-4}$. To evaluate the effect of weight decay, we experiment with AdaBelief using weight decay = $10^{-4}$ and find $\sim 2\%$ drop in top-1 accuracy. So, it may be interesting to see the effect of weight decay on other optimizers.

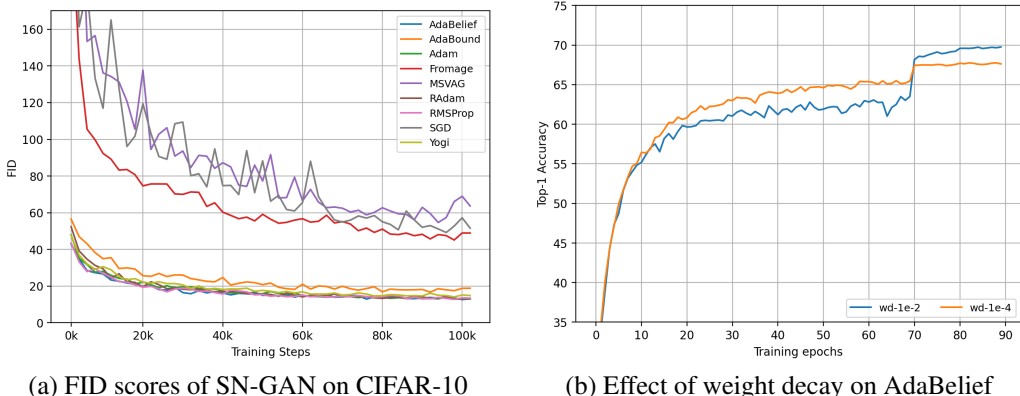

(a) FID scores of SN-GAN on CIFAR-10          (b) Effect of weight decay on AdaBelief

Figure 4: (a) FID values of SN-GAN over training steps for different optimizers (best run plotted out of 5). AdaBelief fares second after RMSProp. (b) AdaBelief performs better when run on larger weight decay of $10^{-2}$.

## 7 Discussion

We now summarize the validity of claims from section 3: (a) Results in section 5.1 show that **AdaBelief outperforms other optimizers in most use cases**. (b) From section 5.2.5, we find that **the convergence speed of AdaBelief is largely in line with adaptive methods**. (c) Based on the analysis in section 5.2.4, we infer that AdaBelief generalizes well, which is evident by its models having lowest bias and relatively low variance. However, it does not uniformly outperform SGD. Therefore, **we fail to completely validate the ability of AdaBelief generalizing as well as SGD**. (d) Even though in section 5.2.3, the least difference between generator and discriminator loss is in case of RMSProp , AdaBelief does outperform other members of the adaptive family. It defeats SGD by a significant margin. Thus, we find that **AdaBelief has stability comparable to adaptive methods in complex settings like GANs**.

**What was easy** The authors provide implementation for most of the experiments presented in the paper. Well documented code and lucid paper helped understand the experiments clearly.

**What was difficult** While hyperparameters (HP) of some experiments were absent (section 5.1.3), some had discrepancies (section 5.1.2). We had to perform grid search for these cases. Training SN-GAN and ImageNet was a resource intensive process which increased the computational burden (Table 1). Formulating the analysis to evaluate the claims of the paper was also challenging 5.2.

**Communication with original authors** We are thankful to the author Juntang Zhuang. He helped us with the implementation and HP details for various experiments. We confirmed the HP for WGAN, SN-GAN, and LSTM experiments. We also clarified the source of Penn Treebank dataset and blacklisting of images in ImageNet.

**Recommendations for reproducibility** Given the time and resource constraints, we performed only a basic analysis of bias-variance trade-off to evaluate the generalization ability of AdaBelief. A more advanced analysis might help in revealing the exact weakness of AdaBelief models in terms of ability to generalize.

Based on our experiments, ablation studies and analysis, we find that AdaBelief is a promising optimizer combining the best of both worlds - accelerated and adaptive gradient methods.

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
