# OpenReview forum: "[Re] AdaBelief Optimizer: Adapting Stepsizes by the Belief in Observed Gradients"
_ML_Reproducibility_Challenge/2021/Fall — RC2021_

### Official Review · Reviewer_NUHU · 2022-03-02
**Reproducibility review**

**Rating:** 7
**Confidence:** 4

**Review:**

The paper discussed clearly the main points of reproducibility challenge. they did extensive ablation study which was one of their strength. Moreover, they provided clear discussions about the results which made it easier to understand them. I think they should have provided recommendations based on the results to help others in improving the reproducibility of their work.

---

### Official Review · Reviewer_Z17s · 2022-03-04
**in detailed and reference provided review**

**Rating:** 10
**Confidence:** 5

**Review:**

starting from the first sentence they already observed all essential and optional parts of the paper, like "scope of reproducibility". Plus they have clearly referenced the paper they are mentioning, using the provided code and methods, hyperparameters, modules, etc. Although they failed to completely validate the scope of AdaBelief generalizing like SGD, they genuinely found notable observations, like the convergence speed of AdaBelief. They fully provided comparing tables in results, which help others to get the concept of paper easily and keep following it.

---

### Meta-Review · Program_Chairs · 2022-04-09

**Recommendation:** Accept
**Confidence:** 5

**Metareview:**

A solid contribution from the authors.  The paper is accepted.

---

### Decision · Program_Chairs · 2022-04-09

**Decision:**

Accept

**Comment:**

Following the recommendation of reviewers and meta-reviewer, the paper is accepted for ML Reproducibility Challenge 2021, and will be published in the upcoming special edition of ReScience Journal.